# Selenium Enrichment Enhances the Quality and Shelf Life of Basil Leaves

**DOI:** 10.3390/plants9060801

**Published:** 2020-06-26

**Authors:** Martina Puccinelli, Beatrice Pezzarossa, Irene Rosellini, Fernando Malorgio

**Affiliations:** 1Department of Agriculture, Food and Environment, University of Pisa, 56124 Pisa, Italy; fernando.malorgio@unipi.it; 2Research Institute on Terrestrial Ecosystems, CNR, 56124 Pisa, Italy; irene.rosellini@cnr.it; 3Interdepartmental Research Center “Nutraceuticals and Food for Health”, University of Pisa, 56124 Pisa, Italy

**Keywords:** biofortification, *Ocimum basilicum*, hydroponic cultivation, antioxidant capacity, ethylene production

## Abstract

The biofortification of leafy vegetables with selenium (Se) is a good way to increase human dietary Se intake. In addition, selenium delays plant senescence by enhancing the antioxidant capacity of plant tissues, decreasing postharvest losses. We investigated the effects of selenium addition on the production and quality of sweet basil (*Ocimum basilicum*) leaves of two harvesting phases, hereafter referred to as cuts, during the crop cycle. Plants were hydroponically grown and treated with 0 (control), 4, 8 and 12 mg Se L^−1^ as selenate. To evaluate the growth, nutritional value and quality of the basil leaves, selected qualitative parameters were determined at harvest and after five days of storage. Application of Se at varying rates (4, 8 and 12 mg L^−1^) was associated with an increased leaf selenium concentration in the first, but not the second cut. The application of Se significantly affected the antioxidant capacity as well as the total phenol and rosmarinic acid contents at harvest. The reduction in ethylene production observed in the plants at 4 mg Se L^−1^ after five days of storage suggests that this Se treatment could be used to prolong and enhance the shelf-life of basil. The daily consumption of 10 g of Se-enriched basil leaves, which, as an example, are contained in a single portion of Italian pesto sauce, would also satisfy the recommended selenium supplementation in humans.

## 1. Introduction

Sweet basil (*Ocimum basilicum* L.) is an annual aromatic herbal plant belonging to the family of Lamiaceae. Due to the high content of essential oils, phenolic compounds, flavonoids [1] and substances with anti-bacterial [2], anti-mycotic [3] and antioxidant activities, basil has pharmacological properties and is used in medicine. Rosmarinic acid is the main phenolic compound found in basil and, together with vitamin E, they have the highest antioxidant activity. Other minor phenolic compounds such as caffeic and ferulic acid are also present in basil leaves [1].

Antioxidant compounds, due to active oxygen species and lipid peroxidation, play an important role in counteracting diseases related to oxidative stress [4]. The leaf content of these compounds, together with the essential oil content, are important parameters to evaluate the quality of basil leaves. As they are highly aromatic, fresh or dried basil leaves can also be used to flavor dishes, and for the preparation of “pesto alla Genovese”, a typical Italian pasta sauce.

Basil is an annual plant native to India and is cultivated all over the world, especially in Asia, Africa and in the Mediterranean area of Europe [5]. Basil is widely cultivated both in open field and under protected cultivation. During the last several years, the hydroponic cultivation of basil has increased. The open field cultivation of basil is characterized by yield variability [6], whereas hydroponic cultivation results in less variability in biomass production, higher yield and quality, and a low use of water and soil [7]. In addition, hydroponically grown plants show a lower level of contamination by pollutants, pests and pathogens, thus reducing losses during the postharvest handling and storage of vegetables [7].

Basil is harvested several times during the growing season by cutting the plants at approximately 15 cm above the ground level in order to allow for re-growth. The detached leaves are highly perishable and have to be preserved against deterioration and spoilage. Little information is available on the postharvest storage, the most appropriate post-harvest processing techniques and the shelf-life of basil leaves [8,9,10].

Selenium (Se) is an essential trace element involved in several biological processes in animals and humans as a component of selenoaminoacids and proteins, and it also has antioxidant functions due to it being a cofactor of glutathione peroxidase [11]. In plants, Se can counteract oxidative stress induced by internal or external factors by improving glutathione peroxidase (GSH-Px) (EC 1.11.1.9) activity [12]. It may also delay plant senescence in several horticultural crops by reducing the ethylene production [13,14,15]. The concentration of Se in non-fortified vegetables is quite low, and does not match the recommended dietary allowance (RDA) for humans of 55 µg Se day^−1^ [16].

Agronomic biofortification with selenium is a good way to increase the Se intake by animals and humans [17], and at the same time to delay senescence and improve the shelf-life, thus decreasing post-harvest losses.

In controlled environments there are two main methods for enriching plants with Se: foliar spraying, and hydroponic cultivation with a nutrient solution containing Se.

In hydroponics, a possible limitation to Se accumulation is when sulfur (S) or phosphorus (P) are added to the nutrient solution. S may affect crop Se uptake because of the antagonistic effect of sulphate on selenate uptake by plants [18]. Increasing the P supply may have negative effects on Se accumulation by inhibiting Se transport [19].

Enhancing the selenium content in basil through foliar spray has been studied more extensively [20,21,22] than biofortification by adding selenium to a nutrient solution [23].

This work investigates the effects of selenium added to a nutrient solution on the growth and quality traits of basil plants grown in hydroponics and cut twice during the growing cycle. Selenium accumulation, yield, and physiological (transpiration rate, ethylene production, net photosynthetic rate) and biochemical (total chlorophyll and carotenoids, total phenols, rosmarinic acid, antioxidant capacity) characteristics were determined in the leaves from two harvesting phases, hereafter referred to as cuts. After five days of storage, the impact of selenium biofortification on selected qualitative parameters (antioxidant capacity, total phenols, rosmarinic acid, total chlorophyll and ethylene production) was evaluated in the basil leaves from the two cuts.

## 2. Results

### 2.1. Climatic Conditions During the Experiment

The climate conditions recorded before the first and the second cuts of basil plants are reported in Table 1. The cumulative radiation, growing degree days and evapotranspiration were generally higher in the period before the first cut than before the second cut.

### 2.2. Effect of Se Application Rate on Se Content, Biomass Production and Quality of Basil

Application of Se in nutrient solution was associated with an increase of Se in the leaves (Figure 1). In the first cut, the leaf selenium concentration was increased by increasing the amount of selenium added. In the second cut, the leaf selenium concentration was lower compared to the first cut, and no differences were detected between the different Se treatments.

The leaf biomass was not affected by the selenium supplementation, and was significantly lower (−36.8%) in the second cut compared to the first cut (Table 2).

The leaf nitrate content significantly increased by increasing the amount of selenium added, and was significantly higher in the second cut (+26.6%) compared to the first cut (Table 2).

On average, stomatal conductance (GS) and net photosynthesis rate (Pn) were lower in plants treated with 12 mg Se L^−1^ compared to control (−26.7% and −41.0%, respectively for GS and Pn) (Table 2). Similar effects on GS, E and Pn were observed in a previous experiment conducted on basil plants grown from February to April and subjected to the same selenium treatments (data not shown). Transpiration rate (E) and Pn were 25.7% and 38.4% lower in the second cut (Table 2).

The Se treatment significantly affected the antioxidant capacity, the total phenol and the rosmarinic acid content at harvest (Table 3). The highest values of the three parameters were detected at 12 mg Se L^−1^. The sample average level of Se concentration induced a higher total phenol content in the second cut (+26.3%) compared to the first cut, with the highest increment in plants treated with 0, 4 and 8 mg Se L^−1^.

The leaf concentration of chlorophylls was not affected by the addition of Se to the nutrient solution, whereas the concentration of chlorophylls was significantly higher in the second cut compared to the first (+39.7% on average) (Table 3).

The leaf concentration of carotenoids was not affected by Se treatments (data not shown).

The leaf ethylene production measured at harvest did not show any differences at 4 and 8 mg Se L^−1^ compared to the control, whereas ethylene increased in plants treated with 12 mg Se L^−1^ in both cuts (Figure 2).

### 2.3. Effects of Selenium on Post-Harvest Quality

After 5 days of storage, treatment with 12 mg Se L^−1^ induced, on average, an increment of leaf antioxidant capacity (+45.8%), total phenol (+36.7%) and rosmarinic acid content (+94.7%) (Table 4).

In the leaves of the second cut, a higher content of total phenols (+21.5% on average) and a lower content of total chlorophylls (−24.7% on average) were detected (Table 4). A reduced ethylene production was detected after five days of storage in plants treated with 4 and 8 mg Se L^−1^ at the first cut (Figure 2B). After five days of storage, plants treated with 12 mg Se L^−1^ showed higher leaf ethylene production compared to 4 and 8 mg Se L^−1^ treatments.

The three-way ANOVA (data not shown) indicated that the ethylene production in control plants after 5 days of storage was significantly higher (*p* < 0.05) than at harvest for both cuts.

## 3. Discussion

This study was conducted to test the effects of selenium addition on the production and quality of sweet basil (*Ocimum basilicum*) leaves. We showed that when different rates of Se were applied, the selenium added to the nutrient solution was absorbed by the roots of basil plants and translocated to the leaves. An increased Se concentration as a result of Se supplementation has been previously observed in basil leaf [23] and in several other species of leafy vegetables, such as lettuce [14,24,25], chicory [14], spinach [26] and chard [27]. The lower amount of Se found in basil tissue in the second cut could be explained by the lower value of cumulative solar radiation observed in the period before the second cut (December) compared to the period before the first cut (November) (Table 1). This may have reduced the evapotranspiration, thus inducing a lower leaf Se accumulation. The uptake of selenate is through a process of active transport and the main way that selenate is translocated is through the xylem, which is closely related to the transport of water. Therefore, evapotranspiration may enhance the uptake and transport of selenium [28]. The Se concentration was also calculated on a fresh weight basis in order to evaluate the effect of the daily consumption of basil leaves enriched with Se (data not shown). The consumption of 10 g of fresh basil treated with 4, 8 and 12 mg Se L^−1^ would provide on average of 33, 84 and 112 µg of Se, respectively. Since the recommended dietary allowance (RDA) for humans is 55 µg Se day^−1^ [16], the daily consumption of 10 g of basil leaves supplemented with 4 mg Se L^−1^ would not reach the toxic threshold, but it would satisfy the rational Se supplementation [16]. On the other hand, 10 g of fresh basil treated with 8 and 12 mg Se L^−1^ would provide a higher amount of selenium than the RDA, but considerably below the toxic threshold (400 µg Se day^−1^).

There were no observable effects of Se application on the biomass of basil. This finding is consistent with studies conducted on lettuce treated with 0.5 and 1.5 mg Se L^−1^ [29], and on spinach treated with 0.21 to 0.41 mg Se L^−1^ [26], which did not show differences in biomass production compared to control plants. However, treatments with 6.2 mg Se L^−1^ have been found to decrease biomass production in cucumber plants [30], thus suggesting that the Se toxic threshold varies depending on the plant species. A lower leaf biomass production was detected in the second cut. This could be ascribed to the reduction in cumulative radiation, growing degree days (GDD), and evapotranspiration detected from the first to the second cuts (Table 1). Horak and Loughin [31] showed that the mean biomass production of *Amaranthus* plants correlated positively with the cumulative radiation and with the GDD.

The increase in nitrate content found in our experiment lies in contrast with the results obtained by Malorgio et al. [14] in lettuce and chicory, where no effects were detected, and with those of Ríos et al. [25], who found a reduction in leaf nitrate content in selenium-enriched lettuce plants. In our experiment, however, the higher nitrate content observed in the Se-treated plants may suggest a slightly toxic effect of Se. An increase in nitrate content had been previously detected in basil plants subjected to abiotic stress, due to the reduction of nitrate reductase activity, by Rai et al. [32]. The lower cumulative radiation in the growth period before the second cut might be responsible for the higher leaf nitrate content detected in leaves of the second cut. Lower radiation can affect the nitrate content by limiting the nitrate reductase activity and, indirectly, by reducing photosynthesis, which leads to a lower reducing power and carbon skeletons needed for amino acid synthesis [33]. The thresholds of nitrates for the commercialization of leafy vegetables based on European regulation 1258/2011 report the maximum values in relation to harvest period and cultivation system for spinach (2000–3000 mg kg^−1^ fresh weight), lettuce (3000–5000 mg kg^−1^, fresh weight) and rocket salad (6000–7000 mg kg^−1^ fresh weight), but not for basil.

The decrease of net photosynthesis rate observed in Se-treated plants can be ascribed to the lower stomatal conductance, since stomata play a key role in regulating the fluxes of water and carbon dioxide between plant and atmosphere. The accumulation of Se in leaf tissues may have also caused damages to the photosynthetic apparatus. There is usually a reduction in photosynthesis in stressed plants [34]. Similar effects on GS, E and Pn were observed in a previous experiment conducted on basil plants grown from February to April and subjected to the same selenium treatments (data not shown). In contrast with our results, wheat plants treated with 2 mg Se-selenate kg^−1^ soil [35] and tomato seedlings treated with 3.95 mg Se-selenite L^−1^ [36] showed an increase in net photosynthesis. The lower evapotranspiration (E) and Pn detected in the second cut may be related to the lower radiation. Radiation is the main climate variable which influences transpiration in protected crops [37], and provides the energy for photosynthesis.

The highest values of antioxidant capacity, total phenol and rosmarinic acid contents detected in the Se-treated plants may be related to the reaction of plants against the potentially toxic effects of Se. Phenols in plant cells form an antioxidant system equivalent to that of ascorbate, which increases the antioxidant capacity and improves the ability of plants to alleviate oxidative stress, and thus counteract the reduction of biomass [38]. An increase in phenolic content following the addition of selenium to the nutrient solution has been observed in tomato plants treated with 25 and 550 µM of Se [39], and in basil plants treated with 3–20 mg Se L^−1^ [21]. Similar results have been reported in basil plants sprayed with 30–120 mg Se L^−1^ [40]. An increased antioxidant capacity and rosmarinic acid content have been previously observed in basil plants treated with 12 mg Se L^−1^ and grown from February to April under the same experimental plan adopted in the present study (data not shown). Basil shoots of the second cut were re-grown after the first cut, and were younger compared to the shoot of the first cut. They showed a higher content of phenols, in accordance with Sgherri et al. [41], who found a higher content of phenols in younger basil plants compared to older plants.

The color of basil leaves was evaluated by monitoring the chlorophyll content. In fact, chlorophyll affects the external quality of the leaves and the color preferences of consumers [42]. Contrasting results have been reported in the literature regarding the effect of Se on chlorophyll concentrations. An increase in chlorophyll content was detected in lettuce and chicory plants treated with 0.5 and 1 mg Se L^−1^ [14] and in wheat plants treated with 2 mg Se kg^−1^ soil [35]. In contrast, a reduction in chlorophyll content was found in lettuce plants [24] when selenate was added to the nutrient solution at doses higher than 3.16 mg Se L^−1^. In fact, because of the similarity with sulfur, Se can substitute S in amino acids, and can thus induce leaf chlorosis as well as a decrease in protein synthesis and dry matter production [43].

The leaf ethylene production measured at harvest was higher in plants treated with 12 mg Se L^−1^ in both cuts. Since ethylene plays an important role in Se resistance in plants [44], this result could indicate a stress or toxic effect of the highest dose of Se used in our experiment.

The higher antioxidant activity and phenol content detected in leaves of plants treated with 12 mg Se L^−1^ after five days of storage are in accordance with evidence of Oraghi Ardebili et al. [40] in sweet basil plants treated with 30–120 mg Se L^−1^ by foliar application. In their experiment, Se-treated plants showed a higher ascorbate and glutathione content and a higher antioxidant activity. The higher phenol content might be due to the effect of Se on the hormonal balance, especially salicylic acid and jasmonic acid, which trigger stress defense responses. There are no data in the literature on the effects of selenium on the rosmarinic acid content in basil during storage.

Various effects of Se on chlorophyll content post-harvest have been detected in chicory and lettuce by Malorgio et al. [14]. In general, the addition of selenium increased the chlorophyll content during storage in both species compared to the control, but the effect was related to the growing season of the plants. In fact, Se supplementation was effective when plants were cultivated in autumn, but not in winter [14].

The higher ethylene production detected in the leaves of the control plants after five days of storage compared to the harvest could be ascribed to the detachment of leaves, which induces senescence and ethylene biosynthesis. The reduction in ethylene production found after five days of storage in plants treated with 4 and 8 mg Se L^−1^ at the first cut is in agreement with results found by Malorgio et al. [14] in lettuce and by Khan et al. [35] in wheat. The reduction in ethylene production could be due to the ability of selenium to be incorporated in the place of sulfur into methionine, leading to Se-methionine [43], which is a precursor of ethylene in the ethylene biosynthesis pathway [45]. Thus, a higher content of Se-methionine in the place of methionine could be the cause of a lower ethylene production [46]. The reduced ethylene production could also be due to the lower activity of the enzymes involved in the ethylene biosynthesis pathway [15]. The enzyme ACC synthase (ACS) catalyzes the conversion of S-adenosyl methionine to ACC; otherwise, the enzyme ACC oxidase (ACO) catalyzes the conversion of ACC to ethylene. Zhu et al. [15] found that Se can downregulate the expression of ACO1, ACS2 and ACS4 in tomato fruit. After five days of storage, plants treated with 12 mg Se L^−1^ showed higher leaf ethylene production compared to 4 and 8 mg Se L^−1^ treatments. This effect may be related to the slightly toxic effects of high Se concentrations [44]. The ethylene accumulated in the packaging during post-harvest storage has been found to induce senescence and quality losses of leaves [47], and thus the reduction in ethylene production could extend the shelf life and improve the post-harvest quality of basil leaves.

Ultimately, the Se concentrations applied in our study make it possible to obtain Se-biofortified basil without inducing a reduction of plant biomass. In general, Se addition did not increase the leaf nitrate concentration above the safety level, and slightly improved the leaf quality parameters. Based on the leaf Se accumulation and on the RDA of Se for humans, 4 mg Se L^−1^ appears to be the optimal concentration for basil biofortification in hydroponics.

## 4. Materials and Methods

### 4.1. Plant Material and Growing Conditions

The experiment was conducted from November 6 through December 21, 2017 at the Department of Agriculture, Food and Environment of the University of Pisa, Italy (lat. 43°40″ N) on basil (*Ocimum basilicum* L. cv. Tigullio) plants. The basil seeds were sown in 240-cell plug trays filled with rock wool and vermiculite, and germinated in a growth chamber at 25 °C for 5 days. Twenty days after sowing, seedlings were transferred to a heated glasshouse and placed into separate hydroponic systems, each consisting of a polystyrene tray floating in a 50 L plastic tank filled with nutrient solution. The nutrient solution contained 12.0 mM N-NO_3_, 1.0 mM P-H_2_PO_4_, 2.44 mM S-SO_4_, 4 mM Ca, 5 mM K, 2 mM Mg, 1 µM Cu, 40 µM Fe, 5 µM Mn, 1 µM Mo and 5 µM Zn. The pH and electrical conductivity (EC) values were respectively 5.6 and 2.04 dSm^−1^, and were checked every 2 days. The nutrient solution was continuously aerated in order to maintain an oxygen content higher than 6.0 g m^−3^. Climatic parameters were continuously monitored by a weather station located inside the greenhouse. During the experiment, the minimum and mean air temperatures were 12.8 °C and 17.5 °C, respectively. The relative humidity was 71.6%. Supplementary lighting was provided by high-pressure sodium lamps (HPS, SON-T 400 W, Philips) for a constant day length of 9 h.

One week after transplanting, selenium, as sodium selenate (Na_2_SeO_4_), was added to the nutrient solution described above at rates of 4, 8 and 12 mg Se L^−1^. To maintain the same Se concentration throughout the experiment, once every two weeks the nutrient solution was replaced with a fresh solution containing the same amount of Se.

The treatments were arranged in a totally randomized design with four replicates, each consisting of a polystyrene tray floating in a 50 L plastic tank filled with nutrient solution. Sixteen plants were planted in each tank; the crop density was approximately 96 plants m^−2^ (on a ground area basis). Basil shoots were harvested on December 4 (1st cut) and December 21 (2nd cut), 21 and 38 days after the selenium supplementation, respectively, when plants were about 30–35 cm tall. The aerial part was cut off at about 10 cm above the soil for fast regrowth, and the second cut was made above the previous cut point to avoid harvesting lignified tissues.

At each sampling, leaves and stem were separated and fresh weight (FW) was determined. The samples were oven dried at 50 °C to constant weight and the dry weight (DW) was recorded.

To investigate the qualitative traits during the post-harvest period, 30 g of fresh basil leaves were put in 1 L plastic trays closed with cling film and stored for five days in the dark at 8 °C and 70–75% RH. Four trays were used for each treatment.

In order to estimate the evapotranspiration, the difference in the volume of the nutrient solution in the 50 L tank between transplanting and the first cut, and between the first and second cuts, were calculated. The difference in the volume was determined by weighing the tank with the nutrient solution and the plants. Since the tanks were almost completely covered by the polystyrene trays, the direct evaporation from the nutrient solution and the water loss due to accidental seepage were negligible. The evapotranspiration of each cut was expressed as L H_2_O m^−2^ of area under cultivation (Table 1).

### 4.2. Chemical Composition Analysis in Basil Leaves

#### 4.2.1. Selenium Analysis

Total selenium content was determined in oven-dried ground leaf samples after digestion with nitric and perchloric acids and reduction by hydrochloric acid [48]. The digests were analyzed by an atomic absorption spectrometer (SpectrAA 240FS, Varian Inc., Mulgrave, Australia) coupled with a hydride generation system (VGA 77, Mulgrave, Varian Inc., Australia). Glass tubes containing only the chemical reagents were used as blanks for the analytical quality controls in order to constantly monitor for Se contamination in the chemical hood.

#### 4.2.2. Nitrate Content

The nitrate content was measured in leaf samples by a spectrophotometer (Perkin-Elmer UV/VIS Lambda 1; Perkin-Elmer, Beaconsfield, Buckinghamshire, UK) using the salicylic-sulfuric acid methods [49]. The nitrate content was calculated using calibration standards containing 0, 1.29, 2.58, 7.741 and 16.127 mmol L^−1^ KNO_3_.

#### 4.2.3. Chlorophyll and Carotenoid Contents

Total chlorophylls, chlorophyll a and b contents, were measured in fresh leaf samples at harvest and after five days of storage using Lichtenthaler’s method [50]. Foliar fresh tissues were cut in small discs and extracted with methanol 99% v/v. The methanol extract was analyzed using a spectrophotometer (Perkin-Elmer UV/VIS Lambda 1; Perkin-Elmer, Beaconsfield, Buckinghamshire, UK) at 662.5, 652.4 and 470 nm. The sample concentrations (µg mL^−1^) of chlorophyll a, chlorophyll b and carotenoids were calculated using the Welburn and Lichtenthaler formula [51].

#### 4.2.4. Total Phenol Content

Total phenol content was measured in fresh leaf samples at harvest and after five days of storage, using the Folin–Ciocalteu reagent [52]. Foliar fresh tissues (0.5 g) were extracted with 99% methanol. For determination of the total phenols content, 100 µL of extract was added to 2 mL distilled water and 300 µL Folin–Ciocalteu reagent. After 4 min, 1.6 mL Na_2_CO_3_ 7.5% (*w*/*w*) was added. After 20 min at 21 °C, the absorbance of samples was read at 765 nm using a spectrophotometer (Perkin-Elmer UV/VIS Lambda 1; Perkin-Elmer, Beaconsfield, Buckinghamshire, UK). The total phenols content was calculated using calibration standards containing 0, 50, 100, 150 and 250 mg gallic acid L^−1^. Values were expressed as g of gallic acid kg^−1^ DW.

#### 4.2.5. Rosmarinic Acid Content

The rosmarinic acid content was measured in fresh leaf samples at harvest, and after five days of storage extracted using ethanol-HCl (ethanol 80%, HCl 1%, H2O 19%). The RA concentration in ethanol-HCl extracts was determined by HPLC analysis and expressed per gram of DW [53]. The HPLC analytical equipment included a Jasco (Tokyo, Japan) PU-2089 four-solvent low-pressure gradient pump and a UV-2077 UV/Vis multichannel detector. Analyses were performed using a Macherey–Nagel C18 250/4.6 Nucleosil^®^ 100–5 column. The detection limit of the analytical method was 0.05 g kg^−1^ DW.

#### 4.2.6. Antioxidant Capacity

The antioxidant capacity of the leaves was measured in fresh samples at harvest and after five days of storage using the ferric reducing ability of a plasma (FRAP) assay [54]. The analysis was performed on the same methanol extracts used for the analysis of the total phenol content. The methanol extract was added to the FRAP reagent and analyzed using a spectrophotometer at 593 nm.

The total antioxidant capacity was determined using the FRAP assay, as adapted by Benzie and Strain [54]. A total of 0.5 g of fresh leaves was extracted using 5 mL of methanol 99% v/v. The samples were then sonicated and maintained at −18 °C for 24 h. Subsequently, 0.1 mL of the methanol extract was added to 0.9 mL of the FRAP reagent, which consisted of 1 mol m−3 2,4,6-tripyridyl-2-triazine (TPTZ) and 2 mol m^−3^ ferric chloride in 250 mol m^−3^ sodium acetate (pH 3.6). This was then mixed and kept at 20 °C for four minutes, and the absorbance was measured at 593 nm using a spectrophotometer (Perkin-Elmer UV/VIS Lambda 1; Perkin-Elmer, Beaconsfield, Buckinghamshire, UK). The antioxidant capacity was calculated using calibration standards containing 0, 50, 100, 150, 250 and 500 mg Fe (II) L^−1^, and the results were expressed as µmol of Fe (II) mg^−1^ DW.

#### 4.2.7. Ethylene Production

Ethylene production was measured in leaves at harvest, and after five days of storage. Leaves were randomly sampled. Two leaves were placed in 85 mL glass tubes (Pirex, France) and closed with holed plastic screw caps fitted with caoutchouc rubber septa. Gas samples (2 mL each) were taken from the headspace of the containers with a hypodermic syringe after 1 h incubation at room temperature. The ethylene concentration in the sample was measured by gas chromatography (HP 5890; Hewlett-Packard, Menlo Park, CA, USA) using a flame ionization detector (FID) and a stainless-steel column (150 cm long × 0.4 cm diameter, packed with Hysep T).

#### 4.2.8. Gas Exchange Measurements

Leaf gas exchange was measured using a CIRAS-2 portable gas exchange system with a broad-leaf chamber, lamp, infrared temperature sensor and automated gas blending system (PPSystems, Haverhill, MA, USA). The photosynthetic active radiation (PAR) and the CO_2_ concentration inside the cuvette were 1000 µmol m^−2^ s^−1^ (above the saturation level) and 400 µmol mol^−1^ air, respectively. There was no active control of leaf temperature and humidity inside the cuvette.

#### 4.2.9. Data Analysis

All data were tested for homogeneity of error variances using Levene’s test, and were subjected to two-way ANOVA with selenium concentration and cut as variables. Data of ethylene production were subjected to three-way ANOVA with selenium concentration, cut and time as variables. Mean values were separated by the least significant difference test (*p* < 0.05). Statistical analysis was performed using Statgraphics Plus 5.1 (Manugistic, Rockville, MD, USA).

## 5. Conclusions

Different growing conditions may affect how basil plants react to selenium supplementation in the nutrient solution. Climate conditions influence evapotranspiration, and consequently the amount of Se taken up by plants and the effects of selenium on the physiology of basil plants.

The addition of 4 mg Se L^−1^ increased the leaf selenium content, without negatively affecting plant growth. The reduction in ethylene production observed in the Se-treated plants after five days of storage suggest that this Se treatment could be used to prolong and enhance the shelf-life of basil. The daily consumption of 10 g of Se-enriched leaves would also satisfy the rational Se supplementation in humans.

The information obtained in our experiment can be utilized to develop a cultivation protocol for the biofortification of basil. Future studies could aim to reveal the best combination of Se concentration used to treat plants, the optimal conditions required to obtain leaves with high content of nutraceutical compounds and high antioxidant activity and the storage conditions.

## Figures and Tables

**Figure 1 plants-09-00801-f001:**
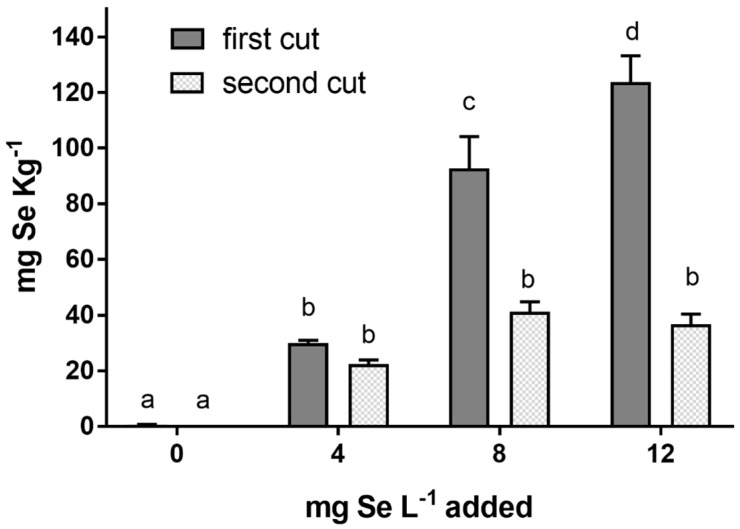
Se concentration (mg Se kg^−1^ dry weight (DW)) in leaves of basil plants subjected to different Se treatments (0, 4, 8 and 12 mg Se L^−1^) and harvested twice from subsequent regrowth (first and second cut). Bars indicated by different letters are significantly different (*p* < 0.05) according to the least significant difference (LSD) test.

**Figure 2 plants-09-00801-f002:**
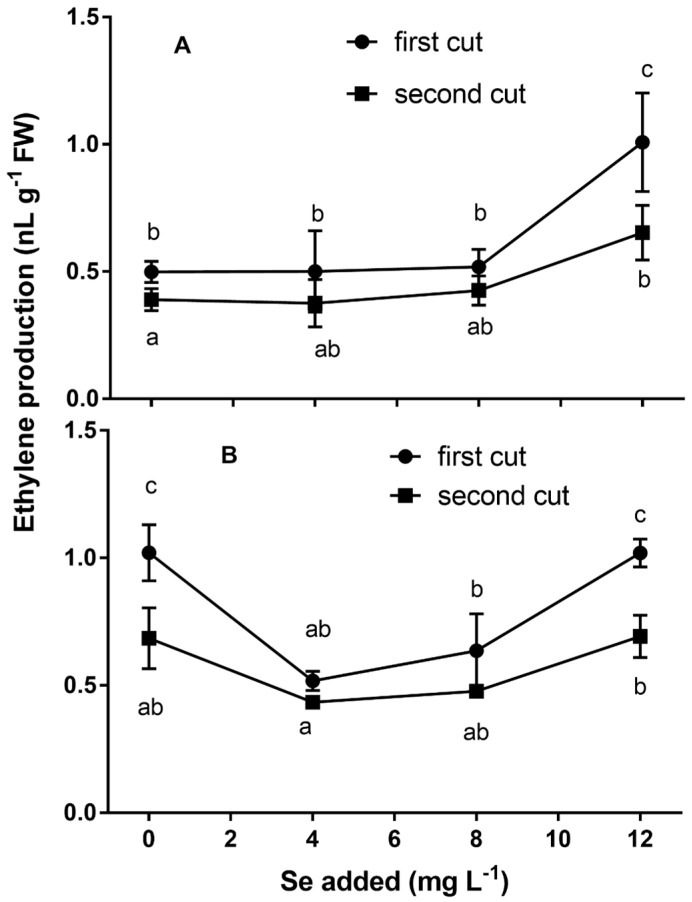
Ethylene production at harvest (**A**) and after 5 days of storage (**B**) in basil plants subjected to different Se treatments (0, 4, 8 and 12 mg Se L^−1^) and harvested twice from subsequent regrowth (first and second cut). Values with different letters are significantly different (*p* < 0.05) according to the LSD test.

**Table 1 plants-09-00801-t001:** Climate conditions recorded in the periods before the first (December 4) and second (December 21) cuts of basil plants.

Climatic Parameters	1st Cut Nov 6–Dec 4	2nd Cut Dec 4–Dec 21
Days of treatment	28	17
Daily mean temperature (°C)	17.5	16.6
Cumulative radiation (MJ m^−2^)	101.5	57.8
Daily mean radiation (MJ day^−1^ m^−2^)	3.5	3.3
Growing degree days (°C day^−1^)	102.9	74.1
Evapotranspiration (L H_2_O m^−2^)	7.54	3.34

**Table 2 plants-09-00801-t002:** Biomass production, nitrate content, transpiration rate (E), stomatal conductance (GS) and net photosynthetic rate (Pn) in leaves of basil plants subjected to different Se treatments (0, 4, 8 and 12 mg Se L^−1^) and harvested twice from subsequent regrowth (1st and 2nd cut). FW = fresh weight; DW = dry weight.

Se Added	Cut	Leaf Biomass	Nitrate Content	E	GS	Pn
mg L^−1^		g DW m^−2^	mg kg^−1^ FW	mmol H_2_O m^−2^ s^−1^	mmol H_2_O m^−2^ s^−1^	mmol CO_2_ m^−2^ s^−1^
0	1st	125 a	2984 a	1.48 a	240 a	8.6 a
	2nd	88 a	3706 a	1.23 a	213 a	4.8 a
4	1st	117 a	3031 a	1.76 a	253 a	9.5 a
	2nd	79 a	3951 a	1.30 a	201 a	5.6 a
8	1st	123 a	3020 a	1.97 a	303 a	9.7 a
	2nd	77 a	4532 a	1.35 a	252 a	5.6 a
12	1st	119 a	3860 a	1.37 a	165 a	4.2 a
	2nd	62 a	4140 a	1.01 a	160 a	3.7 a
Main Effects						
0		107 a	3345 b	1.36 a	226 a	6.7 a
4		98 a	3491 b	1.53 a	227 a	7.5 a
8		100 a	3776 ab	1.66 a	277 a	7.7 a
12		90 a	4000 a	1.19 a	162 b	5.0 b

	1st	121 a	3224 b	1.65 a	240 a	8.0 a
	2nd	76 b	4082 a	1.22 b	206 a	5.4 b
Analysis of Variance						
Cut (A)		***	***	**	ns	***
Se concentration (B)		ns	**	ns	**	**
A × B		ns	ns	ns	ns	ns

Values followed by different letters in the same column differ significantly at 5% level by the LSD test. Significance level: *** *p* ≤ 0.001; ** *p* ≤ 0.01; * *p* ≤ 0.05; ns: not significant. Mean separation within columns by least significant difference (LSD) test, *p* = 0.05.

**Table 3 plants-09-00801-t003:** Antioxidant capacity (µmol Fe(II) mg^−1^ DW), leaf total phenols (mg Gallic Acid Equivalents g^−1^ DW), rosmarinic acid (mg g^−1^ DW) and total chlorophyll (mg g^−1^ DW) at harvest in basil plants subjected to different Se treatments (0, 4, 8 and 12 mg Se L^−1^) and harvested twice from subsequent regrowth (1st and 2nd cut).

Se Added	Cut	Antioxidant Capacity	Total Phenol Content	Rosmarinic Acid Content	Total Chlorophyll
mg L^−1^		µmol Fe (II) mg^−1^ DW	mg GAE g^−1^ DW	mg g^−1^ DW	mg g^−1^ DW
0	1st	236 a	11.1 a	22.3 a	6.9 a
	2nd	248 a	18.9 ab	28.7 a	9.3 a
4	1st	320 a	11.9 a	22.5 a	6.1 a
	2nd	269 a	20.3 b	28.9 a	9.9 a
8	1st	361 a	12.7 a	22.8 a	6.7 a
	2nd	261 a	20.6 b	31.9 a	8.9 a
12	1st	398 a	34.3 c	54.6 a	6.6 a
	2nd	362 a	28.7 c	45.7 a	8.6 a
Main Effects					
0		242 c	15.0 b	25.5 b	8.1 a
4		295 b	16.1 b	25.7 b	8.0 a
8		311 b	16.7 b	27.3 b	7.8 a
12		380a	31.5 a	50.1 a	7.6 a

	1st	329 a	17.5 b	30.5 a	6.6 b
	2nd	285 a	22.1 a	33.8 a	9.2 a
Analysis of Variance					
Cut (A)		ns	***	ns	***
Se concentration (B) (B)		***	***	***	ns
A × B		ns	***	ns	ns

Values followed by different letters in the same column differ significantly at the 5% level by the LSD test. Significance level: *** *p* ≤ 0.001; ** *p* ≤ 0.01; * *p* ≤ 0.05; ns: not significant. Mean separation within columns by LSD test, *p* = 0.05.

**Table 4 plants-09-00801-t004:** Antioxidant capacity (µmol Fe(II) mg^−1^ DW), leaf total phenols (mg GAE g^−1^ DW), rosmarinic acid (mg g^−1^ DW) and total chlorophyll content (mg g^−1^ DW) 5 days after storage in basil plants subjected to different Se treatments (0, 4, 8 and 12 mg Se L^−1^) and harvested twice from subsequent regrowth (1st and 2nd cut).

Se Added	Cut	Antioxidant Capacity	Total Phenol Content	Rosmarinic Acid Content	Total Chlorophyll
mg L^−1^		µmol Fe (II) mg^−1^ DW	mg GAE g^−1^ DW	mg g^−1^ DW	mg g^−1^ DW
0	1st	232 a	12.0 a	24.1 a	7.5 a
	2nd	224 a	14.4 a	25.5 a	5.6 a
4	1st	263 a	11.5 a	33.0 a	6.8 a
	2nd	245 a	16.8 a	34.0 a	5.8 a
8	1st	302 a	12.7 a	30.4 a	7.4 a
	2nd	236 a	16.3 a	30.4 a	5.0 a
12	1st	370 a	21.4 a	66.7 a	7.0 a
	2nd	342 a	22.5 a	47.8 a	5.3 a
Main Effects					
0		228 b	13.2 b	24.8 b	6.6 a
4		254 b	14.1 b	33.5 b	6.3 a
8		269 b	14.5 b	30.4 b	6.2 a
12		356 a	22.0 a	57.2 a	6.1 a

	1st	292 a	14.4 a	38.5 a	7.2 a
	2nd	262 a	17.5 a	34.4 a	5.4 b
Analysis of Variance					
Cut (A)		ns	*	ns	***
Se concentration (B)		***	***	***	ns
A × B		ns	ns	ns	ns

Values followed by different letters in the same column differ significantly at 5% level by the LSD test. Significance level: *** *p* ≤ 0.001; ** *p* ≤ 0.01; * *p* ≤ 0.05; ns: not significant. Mean separation within columns by LSD test, *p* = 0.05.

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
