# Peer review of "Selenium Enrichment Enhances the Quality and Shelf Life of Basil Leaves"

_plants, 2020, doi:10.3390/plants9060801_

Round 1

Reviewer 1 Report

Please consider the following comments:

Lines 10 & 11 – please rephrase. The sentence doesn’t make sense.

Line 13 – mention basil, Ocimum basilicum

Line 14 – use of words “two cuts” is confusing. Do authors mean cut stage or harvesting phase or number of cuttings? It looks like harvesting phases! I think the best way to handle this would be to use the words harvesting phases (hereinafter referred to as cuts).

Line 22 – replace “the” with “a”. Question: what is the daily intake of basil leaves in the target community?

Introduction:

Please provide information about available Se in basil vs recommended Se in human diet.

Table 1 – abolish the unit column and move the units to the end of the parameter e.g., mean temperature (°C)

Line 73 – Rephrase to read, “Selenium was absorbed by basil roots and was translocated to…”. Did you have a marker to show this or is this based on chemical analysis? Would it be safe if this statement was moved to discussion? Here, it would be better to state that application of Se in nutrition solution was associated with an increase in Se on the leaves?

Table 2,3 and 4  - these are very confusing – why were means not compared in some factors? If they did not differ, why not indicate same letter after means in a column? Also, remove the line within the table so that ANOVA and means listed are in the same table. Or separate ANOVA from mean comparison to separate tables – could just put details of the ANOVA in text? If one of the factors is significant, I think it would be better to separate the data. For example, if chlorophyll concentration was significantly influenced by sampling phase, then it means that data should be analyzed separately for each sampling phase (cut).

Discussion:

Lines 138-140 – this paragraph should highlight the overall importance of this study. Why use it to cite what others did, without relating to this study?

Lines 270-272 - Which sentence describes the sampling stage? Is it the days or the heights?

Generally, the discussion section can be shortened by identifying four key points that are interrelated in the study.

Reviewer 2 Report

I read this story with interest. It seems to be carried out well, and it is an elaborate biochemical study. The results are relevant for human health and because it gives insight into plant biological processes.

I have a few minor comments.

First, it seems to me that the Se concentrations used are very high, especially for a hydroponic study. 4-12 mg Se/L as selenate is equivalent to 50-150 micromolar selenate. I would expect that 50 micromolar would already be quite toxic. Are the authors sure about these numbers, and assuming they are, do they have an explanation for why the toxicity was lower in this case than in published experiments? Does the rockwool/vermiculite bind a lot of the Se? I also would expect higher tissue Se levels with these high treatments, e.g. 1000 mg/kg DW for most treatments, but here it was only 40-120 mg/kg DW. If the growth substrate bound most of the Se, wouldn't that create toxic waste (and cost unnecessary Se)?Wouldn't it be advisable, in that case, to use straight hydroponics? Just a point of discussion.

I found the effects on N and on ethylene production interesting to read. Don't S and N tend to correlate, and have positive correlation? Se tends to upregulate S uptake, so maybe that also upregulated N.

I think it would be nice to conclude the discussion with a paragraph summarizing the overall findings, in light of what we have learned about basil's Se responses, and in light of recommendations for basil Se biofortification.

Round 2

Reviewer 1 Report

This paper has improved. Please check the following:

Introduction: please provide some information about the environments where basil are grown and some basic agronomy. How will application of fertilizers and other practices affect the biofortifcation? If the cultivation is in controlled environments, I guess this would be through fertigation or foliar feed? 

Statistical analysis: if a factor is not significant, a mean  comparison would return same letters e.g., 1st cut 14 g A, 2nd cut 14.2 A in the same culumn. Please indicate similar letters for all means that originate from a factor that is not significant. 

Conclusion: please acknowledge the scope of these findings in respect to practical basil production, and how you think biofortification can be implemented.

General: Please read through and ensure that there is good coherency and to correct typos.

Round 3

Reviewer 1 Report

I have gone through the manuscript and I have the following comments:

Title: Could be changed to Selenium application enhances quality and shelf life of basil leaves

Abstract:

Line 10: Delete "the" before "Biofortification"

Line 15: Delete "in order"

Line 19: Begin with "application of Se ..."

Line 18: Begin with "application of Se at varying rates (list) was associated with ....

Introduction:

Lines 65&66: it is not clear what hydroponics cultivation means. Foliar spray or fertigation in a hydroponic system?

Line 70 - use "foliar spray"

Line 82 - Please provide some section headings here. For example: 2.1 Climatic conditions during the life of the experiment - this can give you an opportunity to describe those general aspects that the reader needs to know prior to going into the test results.

2.2 Effect of Se application rate on nutritional content of basil

2.3 Effect of Se application on shelf life of basil

etc. This will make the results section easy to read and understand.

Discussion (this section should be shortened by combining sections that are similar - some paragraphs have only two sentences yet the whole system is related?)

Line 161 - Begin with ... This study was conducted to test...... Here, we report that biofortification of Se in basil through the hydroponic system has a potential to increase the nutritional quality and to improve the shelflife of this important crop. We show that when different rates of Se were applied.... While basil is produced under....., we believe that the findings of this study will contribute to.....

Line 166 - How about you write this section with respect to chronology of results. For example, you can begin by describing why you think harvesting phases differed in each of the evaluated aspects. Then, move to describing whey you think different application rates influenced the evaluated aspects. Generally, do not begin a sentence with "The Se". For example, this sentence would begin with ''A lower amount of Se were found in basil tissue in the second than in the first phases of harvesting... This could have been caused by.... Similar findings were reported earlier in a study conducted in ....and it was postulated that ....

Line 181 - Begin with an opening statement that introduces your finding. For example, There were no observable effects of Se application on the biomass of basil. This finding is in agreement with.... and shows that....

Line 186 - do not highlight and explain a finding in the same sentence, as this is very confusing. Also, begin with a complete highlighting sentence. Example, A lower biomass of basil was observed in the second compared to the first cut. Then, attempt to explain and relate with other studies.

Line 228: Do not begin with "the". How about just begin with Basil.....

Materials and methods

Line 297 - Delete "in order".. begin " To maintain .....

Lines 319 to 390: Create a section called "Chemical composition analysis in basil leaves" then make subsections with each compound that you analyzed. For example, 4.2 Chemical composition analysis of basil leaves . Here, describe general issues about handling of samples and how they were split to allow different analysis of different compounds/nutrients. 4.2.1 Selenium content in leaves

4.2.2 Nitrates in basil leaves. etc. this is just a suggestion. Also, please mention the equipment used, protocol, and appropriate citation for each of them.
